# Physicochemical, Nutritional, Microstructural, Surface and Sensory Properties of a Model High-Protein Bars Intended for Athletes Depending on the Type of Protein and Syrup Used

**DOI:** 10.3390/ijerph19073923

**Published:** 2022-03-25

**Authors:** Jan Małecki, Konrad Terpiłowski, Maciej Nastaj, Bartosz G. Sołowiej

**Affiliations:** 1Department of Dairy Technology and Functional Foods, Faculty of Food Sciences and Biotechnology, University of Life Sciences in Lublin, Skromna 8, 20-704 Lublin, Poland; j.malecki@eurohansa.com.pl (J.M.); maciej.nastaj@up.lublin.pl (M.N.); 2EUROHANSA Sp. z o.o., ul. Letnia 10-14, 87-100 Toruń, Poland; 3Plant in Puławy, ul. Wiślana 8, 24-100 Puławy, Poland; 4Department of Interfacial Phenomena, Institute of Chemical Sciences, Maria Curie-Sklodowska University in Lublin, pl. Maria Curie-Sklodowska 2, 20-031 Lublin, Poland; terpil@poczta.umcs.lublin.pl

**Keywords:** plant protein, liquid fiber, industrial application, nutritional value, optical microscopy, contact angle

## Abstract

The main objective of this study was to investigate the possibility of using a combination of vegetable proteins from soybean (SOY), rice (RPC), and pea (PEA) with liquid syrups: tapioca fiber (TF), oligofructose (OF), and maltitol (ML) in the application of high-protein bars to determine the ability of these ingredients to modify the textural, physicochemical, nutritional, surface properties, microstructure, sensory parameters, and technological suitability. Ten variants of the samples were made, including the control sample made of whey protein concentrate (WPC) in combination with glucose syrup (GS). All combinations used had a positive effect on the hardness reduction of the bars after the storage period. Microstructure and the contact angle showed a large influence on the proteins and syrups used on the features of the manufactured products, primarily on the increased hydrophobicity of the surface of samples made of RPC + ML, SOY + OF, and RPC + TF. The combination of proteins and syrups used significantly reduced the sugar content of the product. Water activity (<0.7), dynamic viscosity (<27 mPas∙g/cm^3^), and sensory analysis (the highest final ratings) showed that bars made of RPC + OF, SOY + OF, and SOY + ML are characterized by a high potential for use in this type of products.

## 1. Introduction

Nowadays, due to the increased interest in this type of product, high-protein bars are important on the food market, primarily in the case of food for athletes and people on vegan and vegetarian diets. Most of the recipes created for scientific research are not directly translated into an industrial scale. Due to the possibility of using a professional industrial line and performing the described tests of high-protein bars on it, the results of our research will be able to be used by food industry plants as an aid in the development of base recipes. Furthermore, based on our research, producers will be able to create high-protein products with the most desirable physicochemical and sensory properties.

Previous studies of high-protein bars available in the literature mainly concerned their tendency to harden over time due to many physicochemical changes in storage, such as Maillard reactions, water activity changes, sugar crystallization, and molecular migrations, which can cause texture hardening. They were also concerned with the method of reducing this tendency by primarily reducing the content of whey proteins in the recipe for their replacement with other commonly used proteins and the use of mainly polyhydric alcohol syrups [1,2].

Nowadays, functional food, including high-protein bars intended for athletes, military food, or ordinary high-protein products available on store shelves, is becoming more and more common. Sports food is not defined in EU law. Until recently, these products could be classified both as foods for particular nutritional uses or as food intended for people exerting intense physical effort, especially athletes. However, due to the change in the regulations on specific food groups, starting 20 July 2016, these products have been defined as general consumption food [3]. Due to a shortage of free time and often feeling overworked, people are more likely to choose ready-made snacks and meal solutions to satisfy their hunger while being reasonably healthy and tasty [4]. In addition, both taste and texture, color and smell also play an important role in making the customer decide to buy a certain product again or not [5]. The increased demand for energy and nutrients requires the consumption of several times more food weight, including protein. With general recommendations that the food consumed by athletes should be of small volume and easy to digest, supplements and functional food become the optimal solution. Their use is more and more common in the world of sports and medicine and among amateurs practicing sports [6,7]. The purpose of consuming these substances (balanced amounts of carbohydrates, fiber, proteins, and fats) is to provide concentrated nutrients that prevent their deficiency in everyday food or increase the absorption of nutrients in an appropriate and harmless way in the body. Moreover, these agents are the source of many bioactive substances (prebiotics, minerals, or unsaturated fatty acids). The use of supplements and functional foods in sports is mainly aimed at accelerating regeneration and increasing body efficiency [8].

Hydrocolloids such as plant protein isolates, currently obtained mainly from soybeans, peas, and rice, which are an alternative to commonly used animal proteins (whey proteins, egg white, albumins, etc.) also exhibit gelling properties during heat treatment. They can be used as emulsion stabilizers, emulsifiers, ingredients to control the crystallization process, thickening ingredients, and foaming, binding, texturing and texture parameters (modification of product hardness and other texture parameters) [9]. Due to the neutral taste and smell, as well as very good functional properties, primarily soy protein isolates are used in the production of foods for special nutritional purposes, sports supplements, people on a diet, pro-health food, milk replacers as well as soups, sauces, mayonnaise, bakery and confectionery products [10]. They are also used as ingredients in edible coatings, mainly in the meat industry. In vegetarian and vegan products, rice and pea proteins are used as enrichment substances in the production of meatless sausages and meat analogues, and more and more often in bakery and confectionery products, which have been enjoying growing interest in the market in recent years [9,11]. Currently, vegetable liquid fibers, particularly oligofructose and a number of innovative syrups containing similar fructooligosaccharides fiber compositions derived from various plants such as corn or tapioca liquid fibers, are among the most commonly used prebiotic substances to replace glucose and glucose-fructose syrups. They exhibit resistance to the action of digestive enzymes in the digestive system. Therefore, they pass into the large intestine, where they are used by the beneficial microflora living in it, influencing its multiplication and improvement of the host’s health significantly [12]. Oligofructoses affect the lipid profile and reduce the level of cholesterol in the blood serum, increase the bioavailability of minerals, prevent and support the treatment of diabetes, have anti-carcinogenic effects and reduce the level of metabolites, allow for the reduction or complete elimination of sugar, glucose syrups and fat in food products, allowing for the formation of attractive dietary products [13]. Polyols, also known as polyalcohols, sugar, or polyhydric alcohols, are a class of semi-synthetic sweeteners. Plums, pears, peaches, apples, olives, figs, strawberries, and raspberries are examples of plants and fruits that contain them naturally. Maltitol, xylitol, sorbitol, lactitol, mannitol, and isomalt are examples of polyols. Sugar alcohols have a lower sweetness than sucrose, allowing them to be employed in larger amounts in food than powerful sweeteners. The lower energy value of these ingredients (2.4 kcal/g) allows for an increase in the percentage of the syrup mass and influences the change of textural parameters, including a reduction in hardness [14,15]. They operate as fillers, like glucose syrups and increase the product’s volume and lowering its specific energy value. They are resistant to enzyme activity and difficult to ferment yet have high chemical stability. Passive diffusion allows polyalcohols to be partly absorbed in the digestive system. Because this process is gradual, it does not result in a surge in blood glucose levels or insulin production by pancreatic cells [16]. However, it should be remembered that the use of polyols in the amount of more than 10% in the finished product requires a declaration on the packaging that the product consumed in large amounts may cause a laxative effect [15].

One of the major problems with high-protein bars is their tendency to harden over time, making the product less affordable and attractive for the consumer. The proteins and syrups used may alter this regularity. This is mainly due to the differences in the origin and individual properties represented by proteins and syrups of plant origin in relation to whey proteins and glucose syrup, most used in the food industry. The mentioned affliction of high-protein bars and the continuous increase in interest in animal protein, high-fructose and glucose syrups equivalents contribute to the growth of this branch of food products. The constant increase in consumer awareness and their search for innovative products require producers to constantly invent new recipes and products [17]. Our previous research was concerned with the effect of various types of proteins on the characteristics and parameters of high-protein bars [18]. The current research focuses on the influence of the best combinations of proteins and syrups on the textural, physicochemical, and sensory parameters in high-protein bars based on selected combinations of soy, rice, pea, and whey proteins, as well as syrups such as oligofructose, liquid tapioca fiber, maltitol, and glucose syrups. The aim of this study was to determine the best possible substitutes for the whey protein concentrate and glucose syrup in the application of high-protein bars made under industrial conditions, considering such aspects as microstructure, water activity, texture analysis, surface tests (contact angle surface), sensory evaluation and a number of physicochemical trials (ultrasonic viscosity, energy, nutritional value, and turbiscan measurement).

## 2. Materials and Methods

### 2.1. Materials

The following ingredients were used in the manufactured products: isolate of soy protein (SOY—proteins ≥87 g/100 g, fat 3.1 g/100 g, ≤1 g/100 g carbohydrates, fragmentation: <200 µm, The Solae Company, Geneva, Switzerland), concentrate of whey protein (WPC—protein ≥80 g/100 g, fat 7.4 g/100 g, carbohydrates 4.1 g/100 g, fragmentation: <200 µm, Polser, Toruń, Poland), concentrate of rice protein (RPC—protein ≥80 g/100 g, fat 1 g/100 g, carbohydrates 6 g/100 g, fragmentation: <300 µm, Barentz, Warsaw, Poland), isolate of pea protein (PEA—protein ≥82 g/100 g, fat 4 g/100 g, carbohydrates 0.8 g/100 g, fragmentation: <200 µm, Cosucra, Warcoing, Belgium), glucose syrup (GS—reducing sugar “DE” 40, viscosity—71.6 Pa∙s, water content—20%, Cargill, Warsaw Poland), oligofructose syrup from chicory (OF—dry matter ≥73–75.5 g/100 g, viscosity—5.0 Pa∙s, water content—25% Cosucra, Warcoing, Belgium), maltitol syrup (ML-maltitol content ≥50 g/100 g, viscosity—4.6 Pa∙s, water content—25% Roquette, Lestrem, France), syrup of tapioca fiber (TF—dry matter ≥75 g/100 g, viscosity—33.0 Pa∙s, water content—25% Anderson Ingredients, Raalte, Holland), rapeseed oil (Zakłady Tłuszczowe Kruszwica, Kruszwica, Poland), maltodextrin (“DE” 15, Amylon, Havlíčkův Brod, Czech Republic), barley malt extract in powder (“EBC-European Brewery Convention” color: 5–12, WES, Wolsztyn, Poland), soy lecithin (Donauchem, Rokietnica, Poland), and natural vanilla flavor (GBD, Warsaw, Poland).

### 2.2. Preparation of High-Protein Bars

The production process was conducted in accordance with the methodology of Małecki et al. [18], as further research related to the topic of high-protein bars. Based on the research carried out so far, the most promising specific combinations of proteins and syrups for this application have been selected. The developed high-protein bars consisted of 38.18 g/100 g protein component (RPC, SOY, WPC, or PEA), 31.82 g/100 g syrup element (OF, GS, ML, or TF), 13.64 g/100 g canola oil, 5.45 g/100 g maltodextrin, 5.45 g/100 g water, 3.64 g/100 g malt extract (from barley), 0.91 g/100 g emulsifier: soy lecithin and 0.91 g/100 g vanilla aroma. The developed products were stored under controlled conditions (relative air humidity 50%, temperature 20 °C) in a plastic container for three weeks.

### 2.3. Texture Profile Analysis (TPA)

The texture attributes were analyzed using the Texture Analyzer TA-XT2i (Stable Micro Systems, Godalming, UK) and Software Texture Expert, as described by Małecki et al. [18]. The measurements were carried out five times. A 36 mm diameter probe (SMS P/36R) was used to doubly press the high-protein bars to achieve 70 percent deformation. The probe motions were interrupted every 5 s, and the test velocity was adjusted at 1 mm/s.

### 2.4. Cutting Strength Test

The cutting test of high-protein bars was performed using a Texture Analyzer (TA-XT2i) in line with the Małecki et al. technique [18]. A Warner Bratzler blade with a slotted reversible blade insert and a blade holder with the knife edge made up the blade set with the knife (HDP/BSK). The knife descended at a rate of 2 mm/s. Five repetitions of the measurements were carried out. The cutting curve was obtained by recording the maximum force the blade needed to cut the sample completely. The results were based on the maximum peak (maximum force) resulting from the shear stress.

### 2.5. Water Activity

On an AWMD-10 water activity meter (NAGY Messsysteme GmbH, Gäufelden, Germany), water activity (a_w_) was measured according to the Małecki et al. method [18]. The measurements were carried out five times at a temperature of 25 °C. For each sample, two outliers were classified as defective and were excluded from further analysis.

### 2.6. Optical Microscopy

The surface and microstructure of the measured high-protein bars were examined using a polarising optical microscope Eclipse E600Pol (Nikon, Tokyo, Japan). The bars samples with an approximate area of one square centimeter were observed directly at the magnifications of ×40, ×100, ×200 and ×400 [19].

### 2.7. Contact Angle Test

Changes of contact angles on the surfaces of the high-protein bars were tested by making use of a contact angle meter GBX (Rue Loire, France), appointed with the digital camera, temperature, and humidity-controlled measuring compartment (20 °C and 50% relative humidity). A droplet (6 μL) from a syringe was put gently on the bar sample surface using the automatic deposition system. The formed contact angle was rated from the droplet shape by the computer software Win Drop. Distilled water was selected for the measurements [20]. The measurements were performed three times for each sample, and average values were calculated.

### 2.8. Turbiscan Measurements

Because syrups are one of the main ingredients in high-protein bars and important in shaping their features, dedicated analyzes were performed for these ingredients.

The changes in the fluidity of syrups used in producing the developed high-protein bars were investigated on the Turbiscan LabExpert fitted with a cooling module—TLab Cooler (Formulation, Toulouse, France) in the 20–60 °C range for 45 min. The processed syrup samples in a glass phial were placed in a temperature-controlled chamber. Then, the collimated light beam (λ = 880 nm) generated by an electroluminescence diode passed through the processed syrup sample, and the transmission detector measured the transmitted light at an angle of 0° while the backscatter detector (using a different diode) recorded the light scattered at an angle of 135°. Based on the analysis the Turbiscan Stability Index (*TSI*) values were determined from the equation (Turbiscan Easy Soft, Formulaction, Toulouse, France) [19]:$$TSI=\sqrt{\frac{\sum_{i=1}^{n}{\left(x^{i}-x_{BS}\right)}^{2}}{n-1}}$$
where *x^i^* is the mean backscatter every 1 min of measurement, *x_BS_* is the mean of *x^i^*, and *n* is the number of scans taken by the instrument.

### 2.9. Ultrasonic Viscosity

The ultrasonic viscometer Unipan 505 type was used to test the dynamic viscosity of high-protein bars (UNIPAN, Warsaw, Poland). The measurements were taken at a temperature of 25 °C. The ultrasound signal level was checked before each measurement. The measuring probe’s tip was entirely immersed in the high-protein bar. The data were measured in mPas∙g/cm^3^ and represented as mPas∙g/cm^3^. All samples were tested three times [18].

### 2.10. Energy and Nutritional Value

The nutritional and energy values of the developed high-protein bars were calculated using the X-mart computer program (X-mart Group, Lublin, Poland) based on the raw material specifications of each ingredient obtained from the individual suppliers. The values were converted into 100 g of the finished product.

### 2.11. Sensory Analysis

The evaluation group consisted of 15 people from EUROHANSA Sp. z o. o. trained in the sensory analysis. The panelists were between 18 and 60 years of age, with no allergies to any of the ingredients in the tested products, and were regular consumers of high-protein products. A five-point scale (1—extremely dislike, 5—extremely like) with the significance coefficients (0.2—color, 0.2—aroma, 0.25—consistency, and 0.35—taste) was used for the study [21,22].

### 2.12. Statistical Processing of the Results

The STATISTICA 13.3 program (Stat Soft Polska, Kraków, Poland) was used to undertake statistical analysis of the acquired findings. The significant differences between the tested samples were assessed using a one-way ANOVA with a Tukey’s post hoc test at *p* < 0.05.

## 3. Results and Discussion

### 3.1. Texture Profile Analysis (TPA), Cutting Test and Optical Microscopy

The effect of the use of various protein-syrup combinations on the texture parameters and the cutting resistance force is presented in Table 1. The microscopic images of the microstructures of the tested high-protein bars are given in Figure 1a–j. Based on the analyzes, significant (*p* < 0.05) differences were observed between the performed trials. According to the research, the control bars made of commonly used sources (WPC + GS) were characterized by the greatest hardness (281.90 N). The least hardness parameters were found in the samples made of soy (18.76 N) and rice (19.92 N) proteins combined with maltitol syrup (SOY + ML and RPC + ML). It is worth noting that least hardness parameters for the individual types of tested high-protein bars are possibly related with the equally small cut resistance parameters.

The hardness assessed by the instrumental methods can be understood as the force required to compress the high-protein bar between the consumer’s thumb and forefinger or as the force required to bite the bar by the molars [23,24]. Fracturability is a parameter that shows how a sample tends to disintegrate during compression [23]. Adhesiveness is the product’s ability to stick to the surface. If the surface of the test sample is sticky, more force will be generated, translating into a feeling of stickiness when eating. Cohesiveness is a reflection of the degree of sample consistency during double squeezing and stretching, instrumentally imitating the chewing process in the mouth [25,26]. The cutting resistance force (shear force) is closely related to the force required to cut the sample by the consumer’s incisors while eating the first bite [23]. The hardness of the high-protein bars is usually due to the great concentration of proteins which, due to such factors as the Maillard reactions, water migration, protein aggregation, or sugar crystallization, may cause hardening during the storage period of this type of product [2,18].

The tested high-protein bars exhibited a variety of texture characteristics. The use of optical microscopy was primarily aimed at showing the differences in the created surface structure depending on the type of protein used and the degree of its fragmentation. A certain regularity was noticed; the high degree of the hardness parameter also reduces the tendency of high-protein bars to stick to the surface. Thus the relatively small adhesiveness and cohesiveness parameters in favor of the usually elevated adhesiveness parameter. This is also confirmed by the Banach et al. research [27]. Based on the Hogan et al. study, hardness, was probably related to the microstructure of the molecules of a given type of protein and their ability to aggregate inside the food product [28]. Based on the research carried out by Małecki et al. [18] and Hogan et al. [28], the differences in the textural parameters and the resistance to cutting of the various types of proteins could be related to pore size and the degree of fragmentation of the protein molecules which changes the degree of moisture migration in the multi-domain products (proteins with smaller pore sizes) and slows down the hardening processes, lowers the cut resistance and an increased tendency to stick to the surface [18,28]. As follows from Figure 1a–j of the microstructure, there are substantial similarities in the size of the pores and the folding of the structure of the tested plant proteins, which probably reduces the TPA parameters and cut-resistance compared to the control sample (WPC + GS). Additionally, a significant effect of the syrups used in the application of the syrups on the reduction of parameters related to the textural analysis and the reduction of resistance to cutting was observed. The explanation for this phenomenon can be the Hassan research (2020), on the basis of which it can be assumed that the use of sugar and glucose syrups alternatives can reduce the degree of high-protein products hardness by reducing the interactions of surface-solvent bonds by reducing the covalent interactions between the proteins and syrups which include hydrogen bonds, van der Waals or ionic forces [29].

### 3.2. Water Activity and Ultrasonic Viscosity

The results of the obtained water activity and ultrasonic viscosity analyses made on the developed high-protein bars are presented in Figure 2a,b. During the storage process, many physicochemical and textural parameters of high-protein products change, mainly in terms of water activity and hardness [30]. For this reason, the developed products were stored under the controlled conditions (relative air humidity 50%, temperature 20 °C) for a period of three weeks. Each bar was packed in a metallized barrier foil and placed in a plastic container. The storage conditions and time were selected on the basis of the tests carried out by Banach et al. [27], which stated that parameters such as water activity change to the greatest extent within a month from the date of production of high-protein products [31].

The differences in the obtained results of the a_w_ and ultrasonic viscosity of the tested products were significant (*p* < 0.05). Water activity determines the course of biological processes, and it especially influences the development of microorganisms. The condition for the growth of microorganisms is that the aw in the environment is maintained at the optimal level for a given microorganism. For most microorganisms, the range is 0.990–0.995. It is assumed that below the water activity value of 0.7, growth and development are not possible for most bacteria and significantly impede the survival of yeast and molds [32]. A certain relationship was found between the water activity and the dynamic viscosity. The PEA + TF high protein bar was characterized by the largest water activity (0.79), and at the same time, it had one of the greatest viscosities (30 mPas∙g/cm^3^). On the other hand, the SOY + OF bar was characterized by the smallest water activity (0.58) and dynamic viscosity (4 mPas∙g/cm^3^). This dependence applied to all tested samples. Based on the previous Małecki et al. [18] studies and the research carried out by Tomczyńska-Mleko et al. [33], differences in the water activity and the ultrasonic viscosity may result from the microstructure of individual types of proteins, their concentration in the product, the tendency to agglomerate them overtime or the amount of air pores in the product formed during the process of mixing and aerating the bar mass. The scientific literature lacks the data regarding ultrasonic viscosity determination, that why it is difficult to find test results to compare with [18,33].

### 3.3. Energy, Nutritional Value and Sensory Evaluation

First of all, for people who pay attention to their diet and control their calories and nutrients, it is important that each snack and meal contain the highest nutritional value and the largest possible content of balanced nutrients [34]. The energy and nutritional value of the tested samples are presented in Figure 3a,b. The obtained results show that in the case of the control sample, the increased energy value was mainly caused by the increased content of carbohydrates, which resulted in the poor balance of individual components. The other trials were characterized by similar energy values and increased protein content, owing to the reduction of carbohydrate content using glucose syrup equivalents. For athletes, the diet should promote the development of exercise capacity and rapid regeneration after large physical loads. The qualitative composition of meals, their distribution, and energy value should be related to the size of energy losses and the metabolism characteristic of training loads. Provided in the right proportions, proteins, fats, and carbohydrates are the sources of ATP, a high-energy compound that breaks down into ADP during muscle work with the simultaneous release of energy. Their number and properly selected proportions must stimulate the appropriate energy dosage depending on the type of practiced sports discipline, the duration of exercise, and changes in its intensity [35]. All the tests carried out fit in this trend.

However, it should be noted that the product made of SOY + ML was characterized by the greatest degree of energy value reduction. This is probably due to the reduced caloric content of maltitol (2.4 kcal/g). It is worth paying special attention to the dietary fiber content in the obtained samples, especially RPC + TF (26 g/100 g), PEA + TF (23 g/100 g), and SOY + TF (23 g/100 g), which had the highest levels of fiber content. The products to which liquid fibers were added during the production process are characterized by a high fiber content in the final product and may be an additional selection criterion for potential consumers. Fiber products also allow the claim to be “high in fiber” in line with the current EU legislation [36]. In addition, dietary fiber is an important bioactive component that plays a significant role in rational nutrition, as well as in the treatment and prevention of many diseases. Its action mainly involves regulating intestinal peristalsis, preventing constipation, removing toxins and metabolic products from the body, and consequently reducing the risk of cancer, especially of the large intestine. An important property of insoluble fiber is the ability to bind carcinogenic, mutagenic, and other toxins formed during the digestion of food. The fiber that binds to the toxins is eliminated from the body in the stool. The soluble fibers can be broken down into short-chain fatty acids such as butyrate, propionate, and acetate by fermentation. The addition of fiber to the food products reduces the energy density of food and also extends the time of feeling full [37,38,39].

The sensory analysis showed that the high-protein bars made of soy and rice proteins enjoyed the highest ratings, with particular emphasis on the repetitions containing mainly oligofructose and tapioca fiber (RPC + OF, SOY + OF, and SOY + TF) (Figure 4). The sample made of soy protein and, with the addition of polyhydric alcohol (maltitol), was also highly rated (SOY + ML). High scores of these tests can be linked with the relatively small parameters of ultrasonic viscosity and the parameters of texture analysis of the same products, which could contribute significantly to the positive assessments of the respondents. The research results [18] show that the bars made of rice and soy proteins were highly rated during the sensory evaluation, which is also confirmed by the analyses. On the other hand, according to the Gunyaphan et al. studies (2020), the protein bars made of pea proteins were assessed very positively by consumers. The differences in these results may result from the concentration of proteins in the finished product and the degree of their fragmentation and deodorization [40].

Considering the high-protein products available on the store shelves, they are often coated in chocolate. It should be remembered that mainly dark chocolates (with a high cocoa mass content) have the ability to mask the aftertaste of proteins, so it can be assumed that pouring the tests in chocolate would increase the ratings of each high-protein product [41,42].

### 3.4. Water Droplets Kinetics—Contact Angle

Wettability is a very important physical property that characterizes the surface of materials. Its value determines the basic functions of food products and other materials, including adhesion or lubricity [43]. The purpose of the study of the surface structure was to determine the degree of hydrophobicity, hydrophilicity, and surface roughness depending on the type of protein and syrup used. The measure of the wettability of a solid surface with a small molecular weight liquid is the angle (colloquially called the contact angle) between the tangent to the droplet at the point of contact with the surface being tested and this surface. Thus, the measurement of the static contact angle can be reduced to placing a drop of a low molecular weight liquid on the surface of the tested material, measuring the angle of inclination of the tangent to the outline of the drop surface at the point of its contact with the substrate [44]. The results of the obtained contact angles are shown in Figure 5. They indicate significant differences in the obtained parameters depending on the combination of proteins and syrups.

The sample consisting of RPC + ML was characterized by the greatest surface hydrophobicity as the contact angles were the highest (close to 80°). The lowest contact angles were characterized by the product made of PEA + TF, so the surface of this product showed the most hydrophilic features. It is also worth noting that in the case of this test, this angle decreased very quickly over time. Based on the research of Perez-Huertas et al. (2020) and Huhtamaki et al. (2018), it can be assumed that the differences in the contact angles can be influenced by the roughness of the given surface of the tested products resulting from the substances that make it up [20,45]. The tested bars were characterized by the ability to spread water droplets quickly over the surface compared to the products of other researchers. Considering the analyses, this may be related to the relatively small water activity of the presented products and the large content of dry ingredients with great water absorption (proteins, barley malt, maltodextrin), which is also confirmed by the Ojogbo et al. studies, in which the products with a dry structure were tested. It is worth noting that the obtained contact angle results are usually not equal to 0, which probably indicates a large degree of surface roughness of the tested products [44]. On the other hand, based on the Małecki et al. research, it can also be assumed that the degree of moisture absorption by a given type of proteins can be related to the different microstructure of individual raw materials, in particular proteins (the degree of their fragmentation and the ability to create conglomerates) [18].

### 3.5. Turbiscan Measurements

Turbiscan measurements enable fast and sensitive identification of destabilization mechanisms (such as creaming, sedimentation, etc.). A temperature-controlled measurement cell allows stability monitoring at specific storage temperatures or accelerates the destabilization process. The obtained Turbiscan Stability Index (*TSI*) results for the individual types of syrups used to produce the tested high-protein bars are presented in Figure 6. The Turbiscan Stability Index measures the global stability of a product and is used to compare the stability of different samples. Higher *TSI* values mean greater system instability. Accordingly, the *TSI* parameter is called the instability factor [46]. Worthily to this methodology, the TF syrup was characterized by the greatest stability in the full scope of the study. The most unstable sample was GS, commonly used in the food industry. Based on the Schellart research (2011), the great instability of this syrup may result from its very high ability to change viscosity depending on the temperature. At a low temperature, glucose syrups are characterized by very large viscosities, and with increasing temperature, their viscosity decreases significantly [47]. This is critical during the syrup dispensing process for various types of products. Typically, the syrup is heated to provide an easier pumping process and to reduce the load on the pumps. Taking into account the research carried out by Małecki et al., it can be assumed that such a great GS viscosity at temperatures close to 20–30 °C may be the reason for the increased hardness of high-protein products, which is undesirable [18].

In turn, the TF syrup was characterized by large fluidity and little viscosity in the entire measuring range. Taking into account other analyses performed, a small *TSI* for the TF syrup may be the reason for the tendency of high-protein bars towards the higher water activity results and accelerated risk of surface drying due to the moisture migration into the internal structures and increasing the water absorption rate and brittleness of the surface. Taking into account the *TSI* results obtained for the OF and ML syrups, they were characterized by balanced *TSI* ranges. According to the Nastaj et al. (2020) studies, also the type and concentration of protein may have an effect on the *TSI* parameter [19].

## 4. Conclusions

High-protein bars made of a combination of protein components and syrup substances selected based on the previous studies showed significant differences in the tests. It should be noted that all the presented solutions resulted in the desired lowering the degree of hardness in relation to that of the control sample, as the tendency towards hardening over time is one of the most important parameters regarding this type of product. Each of the proteins used had a different microstructure, as evidenced by the images from an optical microscope. Based on the current and previously carried out research, it can be assumed that this had an impact on the variable characteristics of each type of bar, primarily water activity and dynamic viscosity, where the best results were the bars made of soy proteins (SOY) and rice proteins (RPC) in combination with the OF and ML syrups. In terms of nutritional and energy values, all trials were on a similar level where there was a significant decrease in the contents of carbohydrates and sugars and an increase in the percentage of protein compared to the control sample, which may be an additional attractive feature for athletes and people who care about a low carbohydrate diet. Taking into account the analysis of contact angles, it can be assumed that the combinations of SOY and RPC proteins with OF and ML syrups ensure the hydrophobicity of the product surface similar to standard products made with WPC + GS. The *TSI* parameter suggests that the glucose syrup (GS) remains the least attractive syrup in terms of the production of this type of product due to a very wide spectrum of viscosity changes in different temperature ranges, which decreases the scope of its application. On the other hand, the remaining syrups showed little differentiation in *TSI*, which proves their stability and slight changes in viscosity in various temperature ranges, which may be beneficial for the application of this type of high-protein bars. It can be assumed that SOY + OF, SOY + ML, and RPC + OF may be the best alternatives to the commonly used WPC + GS, owing to the attractive assessments and parameters in practically each of the analyses being made. The combinations of the syrups with the pea proteins also deserve attention due to the significant reduction in hardness and energy value of the obtained products.

## Figures and Tables

**Figure 1 ijerph-19-03923-f001:**
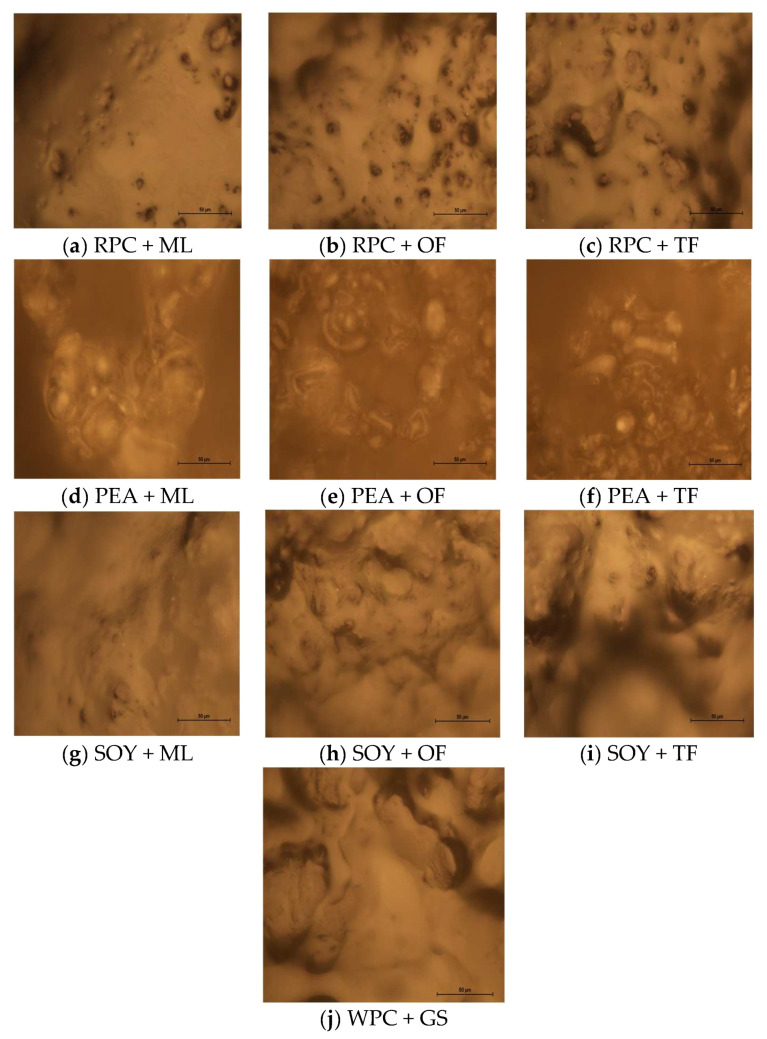
(**a**–**j**) Microstructure of the surface of the tested high-protein bars from the optical microscope (MAG: 400×). The method used to present photos of the microstructure of the surface for comparison for individual components.

**Figure 2 ijerph-19-03923-f002:**
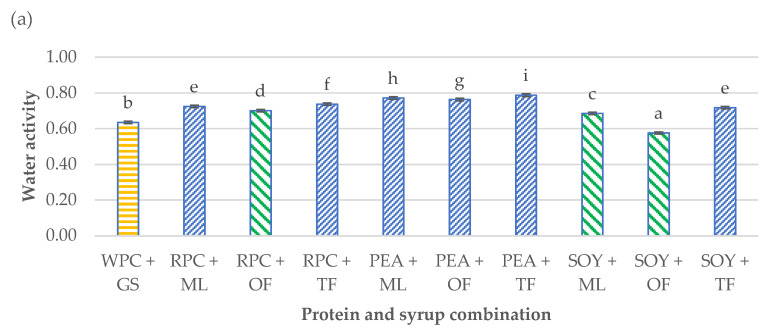
Influence of the combination of proteins and syrups on the (**a**) water activity (a_w_) and (**b**) ultrasonic viscosity of the developed high-protein bars. The letters (a–i) indicate significant differences at *p* < 0.05 (Tukey’s HSD test). The control sample color is yellow. The best prognostic samples for both determinations are marked green. The a_w_ tests were carried out in five repetitions (*n* = 5) and ultrasonic viscosity in three replications (*n* = 3).

**Figure 3 ijerph-19-03923-f003:**
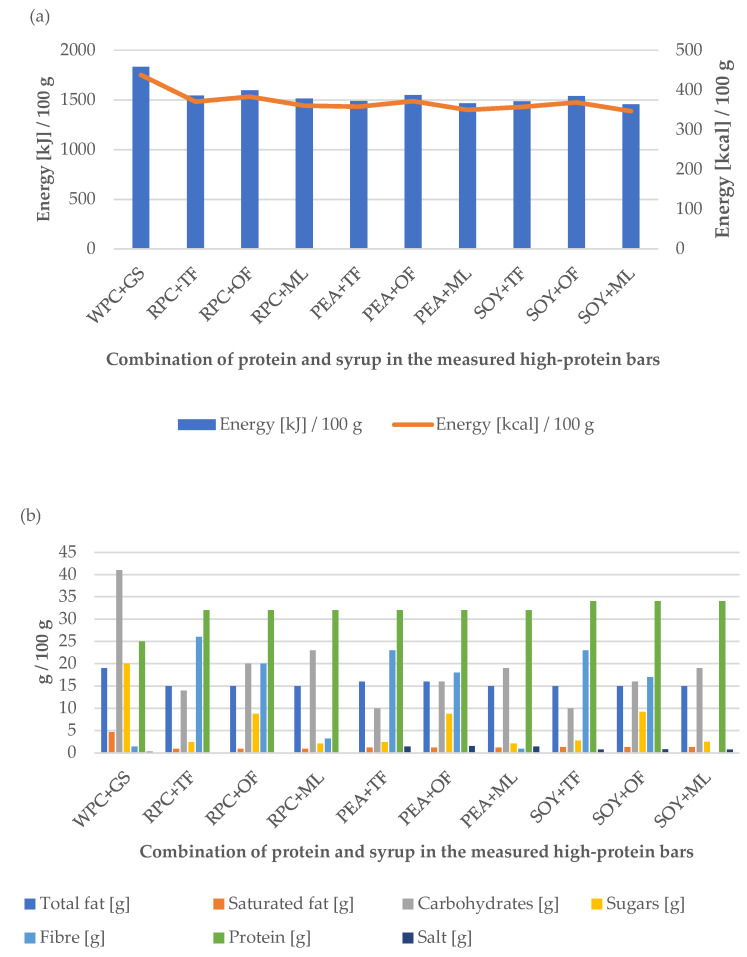
Influence of the combination of proteins and syrups on the (**a**) energy value and (**b**) nutritional value of the developed high-protein bars.

**Figure 4 ijerph-19-03923-f004:**
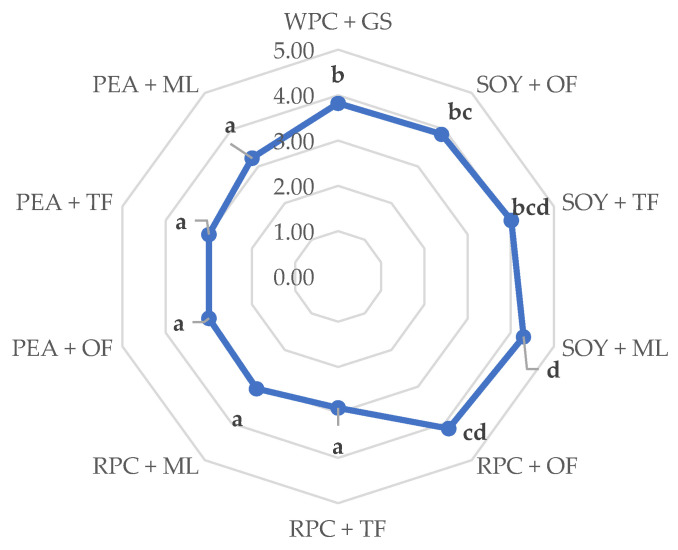
Influence of the combination of proteins and syrups on the sensory evaluation of the developed high-protein bars. The letters (a–d) indicate significant differences at *p* < 0.05 (Tukey’s HSD test). Fifteen trained evaluators participated in the study (*n* = 15).

**Figure 5 ijerph-19-03923-f005:**
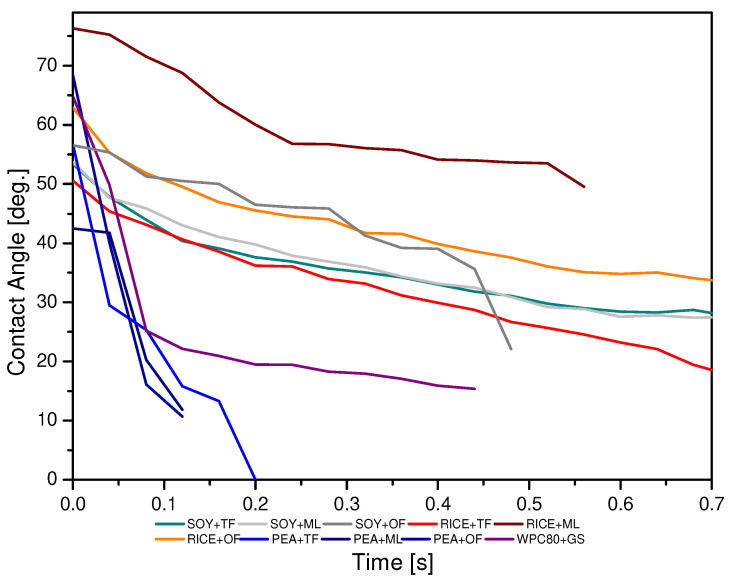
Comparison of individual water droplets kinetics depending on the type of protein and syrup used in the production of the developed high-protein bars.

**Figure 6 ijerph-19-03923-f006:**
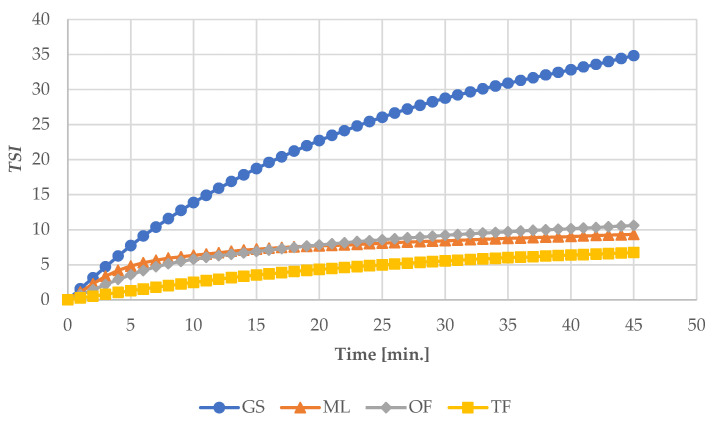
Changes in the *TSI* over time during the heating of individual syrups used in the production of high-protein bars being the subject of implementation.

**Table 1 ijerph-19-03923-t001:** Impact of different protein and syrup combinations on the high-protein bars texture attributes and cutting resistance.

Combination of Protein and Syrup	Texture Attributes				Cutting Resistance Force [N]
	Hardness [N]	Fracturability [N]	Adhesiveness [J]	Cohesiveness	
WPC + GS	281.90 ^h^ ± 1.32	0.32 ^ab^ ± 0.13	1.66 ^f^ ± 0.11	0.28 ^h^ ± 0.01	49.27 ^i^ ± 0.15
RPC + ML	19.92 ^a^ ± 0.48	0.16 ^a^ ± 0.02	1.53 ^e^ ± 0.05	0.14 ^e^ ± 0.01	8.27 ^a^ ± 0.03
RPC + OF	27.67 ^b^ ± 0.18	0.27 ^ab^ ± 0.02	3.34 ^g^ ± 0.03	0.12 ^de^ ± 0.01	11.51 ^b^ ± 0.14
RPC + TF	35.53 ^c^ ± 0.49	0.47 ^ab^ ± 0.02	0.10 ^a^ ± 0.01	0.10 ^cd^ ± 0.01	15.73 ^d^ ± 0.16
PEA + ML	56.22 ^d^ ± 0.30	89.13 ^c^ ± 0.43	0.14 ^a^ ± 0.01	0.02 ^a^ ± 0.01	29.79 ^f^ ± 0.05
PEA + OF	106.68 ^f^ ± 0.22	141.45 ^d^ ± 0.79	0.17 ^ab^ ± 0.01	0.07 ^bc^ ± 0.01	48.55 ^h^ ± 0.23
PEA + TF	71.76 ^e^ ± 0.29	157.33 ^e^ ± 1.08	0.25 ^b^ ± 0.03	0.06 ^b^ ± 0.01	49.94 ^j^ ± 0.07
SOY + ML	18.76 ^a^ ± 0.62	0.13 ^a^ ± 0.01	1.51 ^e^ ± 0.03	0.25 ^gh^ ± 0.03	14.26 ^c^ ± 0.15
SOY + OF	136.46 ^g^ ± 2.97	1.14 ^b^ ± 0.10	1.21 ^d^ ± 0.06	0.24 ^fg^ ± 0.01	46.58 ^g^ ± 0.21
SOY + TF	34.82 ^c^ ± 0.65	0.30 ^ab^ ± 0.02	0.48 ^c^ ± 0.01	0.23 ^f^ ± 0.02	21.53 ^e^ ± 0.21

The data are presented as means ± SD (standard deviation). ^a–j^ Means in the same column with different superscripts are significantly different (*p* < 0.05, Tukey’s honest significant difference “HSD” test). The tests were carried out in five replications (*n* = 5).

## Data Availability

The data presented in this study are available on request from the corresponding author.

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
