# Peer review of "Physicochemical, Nutritional, Microstructural, Surface and Sensory Properties of a Model High-Protein Bars Intended for Athletes Depending on the Type of Protein and Syrup Used"

_ijerph, 2022, doi:10.3390/ijerph19073923_

Round 1
Reviewer 1 Report
The manuscript titled:” Physicochemical, Nutritional, Microstructural, Surface and Sensory Properties of a Model High-Protein Bars Intended for Athletes Depending on the Type of Protein and Syrup Used” brings interesting data about the protein bars. However, for the purpose of publication in the journal there are needed improvements and corrections.
Introduction needs to be improved. Authors starts with functional foods and consumers expectations, than refers to recommendation, and describe athletics needs. Next paragraph starts with description with chemical substances. I would suggest to start with substantiation why the protein bars are of the authors interest, than describe the current status of scientific knowledge of these products,, with listing the main substances. Authors, also should take into the consideration legal requirements for the composition of foods for particular nutritional uses (regulated by law).
Line 24. “changes 33 in storage, such as Maillard reactions, water activity changes,”. The Maillard reaction - the reaction takes place at high temperatures so how it can take place during storage? How during storage of bars (always in foil cover, that protect form water losses and oxidization that changes in flavor as a total) water activity can change?
Lines 45-48. This sentence is obvious and should not be included here.
Lines 50-54. Joining together the need of small volume of the meal and its easiness to digest in the aspect of two different group of consumers: athletes, convalescents (with completely different demanding) is misunderstanding. Especially bars should not be used as an example of the possibility of replacing a whole meal (due to the high degree of processing). Except of the examples of foods for particular medical use.
Lines 56 and 58. What kind of SUBSTANCES or AGENTS?
Lines 90-91. “Sugar alcohols have much smaller sweetness than sucrose, so they can be used in food in greater amounts than intense sweeteners”. This sentence sound as a gain. What about previous paragraphs where authors claim that due to athletics higher demanding for food (also in a volume) it is in need do provide highest nutrient in smaller volume. Also, what with laxative effect of high amount of polyols in the diet.
Lines 116-118:
-Microstructure- authors should describe the need of this examination as well as explain (better) the photos (figure 1).
-Microbiological safety- it cannot be defined only by water activity. These words need to be change.
-Surface tests- what is the aim and need of this test?
- “a number of other physicochemical tests”-do authors mean nutritional value calculation? If yes it should be written clearly, if not what others? also should be pointed
Subparagraph 2.4. What “method (16) using”? please explain
Subparagraph 2.5. What “method (16) on”? please explain
What for authors were using optical microscope? It wasn’t the method used to obtain results, only to present the pictures for comparison. I would suggest just to put this information below the photos.
Subparagraph 2.8 Why authors were conducting this measurement since manuscript is about bars and their properties? If the measurements are about the syrup also, its viscosity , water content should be included especially in the frame of changing this parameters dependently form technological temperature used.
Subparagraph 2.10- converted or expressed?
Line 297. The numbers are not correct, these values are valid for bacteria growth, for yeast and mold average water activity is 0,8 and 0,7 (respectively).
Lines 322-324. Why authors put here sentence about biochemistry of human cells and ATP transformation?
How authors refer to the fact that their bars had higher energetic value (370-410 kcal/100g) than most common on the market (250-350 kcal/100g)?
Line 332-333. this sentence is not necessary, the polyols are not the fiber.
Line 343. In which way fiber is removing toxins in human body?
Subparagraphs 3.4 and 3.5. There is lack of explanation what for such measurements were conducted, the aim for it.
Line 454. “syrups provide hydrophobic properties similar to those of the standard products.” And what are these standard hydrophobic properties?
There is very few discussion (comparison with other authors) as well as poor explanation of some conducted measurements.
Very often authors are writing “method by [number]”, please correct.
I suggest major revision.
Author Response
We have corrected our manuscript with regard to Reviewers comments:
Reviewer #1: yellow color
Reviewer #2: green color
Reviewer #3: blue color
If Reviewers had the same comments: pink color
Editorial Office comments: grey color
Reviewer #1: Review of Manuscript: ijerph-1613239
Reviewer 1
The manuscript titled:” Physicochemical, Nutritional, Microstructural, Surface and Sensory Properties of a Model High-Protein Bars Intended for Athletes Depending on the Type of Protein and Syrup Used” brings interesting data about the protein bars. However, for the purpose of publication in the journal there are needed improvements and corrections.
1) Introduction needs to be improved. Authors starts with functional foods and consumers expectations, than refers to recommendation, and describe athletics needs. Next paragraph starts with description with chemical substances. I would suggest to start with substantiation why the protein bars are of the authors interest, than describe the current status of scientific knowledge of these products,, with listing the main substances. Authors, also should take into the consideration legal requirements for the composition of foods for particular nutritional uses (regulated by law).
Lines: 32-40; 50-54; 58-67; 100-103; 108-110; 126-132
Thank you for your comment. An initial fragment has been added to present the reason for our interest in the subject of high-protein bars. Clarification is also provided on the definition of food for particular sports nutrition in accordance with the applicable EU law.
2) Line 24. “changes 33 in storage, such as Maillard reactions, water activity changes,”. The Maillard reaction - the reaction takes place at high temperatures so how it can take place during storage? How during storage of bars (always in foil cover, that protect form water losses and oxidization that changes in flavor as a total) water activity can change?
Thank you for your comment.
- a) Referring to the publications below, the Maillard reaction may take place in such products during the storage process, but with a lower intensity than in the case of thermal processes.
“If a food matrix contains both proteins and reducing sugars, the Maillard reaction would occur, which could result in the glycation of protein molecules and even the aggregation of proteins (Chevalier and others 2001)1. Given the fact that the Maillard reaction develops fast in the water activity of 0.6 to 0.8 (Labuza and Saltmarch 1981)2, the properties of intermediate-moisture foods such as nutritional bars containing reducing sugars and proteins may suffer from the Maillard reaction during storage.”
1 Chevalier F, Chober JM, Molle D, Haertle T. 2001. Maillard glycation of beta-lactoglobulin with several sugars: comparative study of the properties of the obtained polymers and of the substituted sites. Lait 81:651–66.
2 Labuza TP, Saltmarch M. 1981. The nonenzymatic browning reaction as affected by water in foods. In: Rockland L, editor. Water in foods. New York: Academic Press. p 605–50.
- b) According to the available publications, in the case of protein bars it does not matter whether they are packed or not. Moisture migration and changes in water activity are happening anyway. Possibly with varying intensity.
“As an important component in foods, the content and distribution of water have a significant influence on the quality characteristics and storage stabilities of foods (Danshi Zhu et al., 2017)1. In the early stage of storage, and there was a water activity gradient between its components (Mezzenga, 2007)2. Driven by the difference in potential energy, moisture and other small molecules migrated from the high-moisture-activity region to the low-moisture-activity region (Purwanti, Goot, Boom, & Vereijken, 2010; Roudaut & Debeaufort, 2010)3,4. This migration would allow water to diffuse from proteins to sugars and glycerol, causing water-cooperative changes in the protein molecules, leading to an overall hardening texture of the HPNBs (high protein nutrition bars) (Loveday et al., 2010)5. Meanwhile, due to water transfer, the plasticizing ability of the HPNBs was reduced, leading to further hardening (Y. Li, Szlachetka, Chen, Lin, & Ruan, 2008)6.”
1 Zhu, D., Liang, J., Liu, H., Cao, X., Ge, Y., & Li, J. (2017). Sweet cherry softening accompanied with moisture migration and loss during low temperature storage. Journal of the Science of Food and Agriculture, 98(10), 3651–3658. https://doi.org/ 10.1002/jsfa.8843
2 Mezzenga, R. (2007). Equilibrium and non-equilibrium structures in complex food systems. Food Hydrocolloids, 21(5), 674–682. https://doi.org/10.1016/j. foodhyd.2006.08.019
3 Purwanti, N., Goot, A. J. V. D., Boom, R., & Vereijken, J. (2010). New directions towards structure formation and stability of protein-rich foods from globular proteins. Trends in Food Science & Technology, 21(2), 85–94. https://doi.org/10.1016/j. tifs.2009.10.009
4 Roudaut, G., & Debeaufort, F. (2010). Moisture loss, gain and migration in foods and its impact on food quality. Chemical Deterioration and Physical Instability of Food and Beverages, 143–185. https://doi.org/10.1533/9781845699260.2.143
5 Loveday, S. M., Hindmarsh, r. P., Creamer, r. K., & Singh, r. (2010). Physicochemical changes in intermediate-moisture protein bars made with whey protein or calcium caseinate. Food Research International, 43(5), 1321–1328. https://doi.org/10.1016/j. foodres.2010.03.013
6 Li, Y., Szlachetka, K., Chen, P., Lin, X., & Ruan, R. (2008). Ingredient characterization and hardening of high-protein food bars: An NMR state diagram approach. Cereal Chemistry, 85(6), 780–786. https://doi.org/10.1094/CCHEM-85-6-0780
3) Lines 45-48. This sentence is obvious and should not be included here.
Thank you for your comment. The fragment has been removed.
4) Lines 50-54. Joining together the need of small volume of the meal and its easiness to digest in the aspect of two different group of consumers: athletes, convalescents (with completely different demanding) is misunderstanding. Especially bars should not be used as an example of the possibility of replacing a whole meal (due to the high degree of processing). Except of the examples of foods for particular medical use.
Lines: 58-67
Thank you for your comment. The fragment has been changed.
5) Lines 56 and 58. What kind of SUBSTANCES or AGENTS?
Lines: 63-67
Thank you for your comment. Information on substance examples has been added.
6) Lines 90-91. “Sugar alcohols have much smaller sweetness than sucrose, so they can be used in food in greater amounts than intense sweeteners”. This sentence sound as a gain. What about previous paragraphs where authors claim that due to athletics higher demanding for food (also in a volume) it is in need do provide highest nutrient in smaller volume. Also, what with laxative effect of high amount of polyols in the diet.
Lines: 100-103; 108-110
Thank you for your comment. We wanted the possibility of using more of this type of ingredients in the form of a syrup while reducing the energy value or keeping it at a similar level. A higher percentage of syrup ingredients also reduces parameters such as hardening in the case of high-protein bars. Suggested information has been added.
7) Lines 116-118:
-Microstructure- authors should describe the need of this examination as well as explain (better) the photos (figure 1).
Lines: 275-277
Thank you for your comment. Information has been added regarding the purpose of examining the microstructure by optical microscopy.
-Microbiological safety- it cannot be defined only by water activity. These words need to be change.
Lines: 324-325
Thank you for your comment. The microbiological safety information for testing with water activity has been changed.
-Surface tests- what is the aim and need of this test?
Lines: 406-408
Thank you for your comment. Information on the purpose of the surface tests has been added.
- “a number of other physicochemical tests”-do authors mean nutritional value calculation? If yes it should be written clearly, if not what others? also should be pointed
Lines: 126-132
Thank you for your comment. Information on other tested parameters has been added. The explanation of the purpose of a given type of research is provided at the beginning of each of the described subsections, depending on the research being described.
Subparagraph 2.4. What “method (16) using”? please explain
Lines: 176-178
Thank you for your comment. Information on the method used has been added. The sections explain what the method is based on.
Subparagraph 2.5. What “method (16) on”? please explain
Lines: 182-183
Thank you for your comment. Information on the method used has been added. The sections explain what the method is based on.
8) What for authors were using optical microscope? It wasn’t the method used to obtain results, only to present the pictures for comparison. I would suggest just to put this information below the photos.
Lines: 272-273
Thank you for your comment. Suggested information has been added.
9) Subparagraph 2.8 Why authors were conducting this measurement since manuscript is about bars and their properties? If the measurements are about the syrup also, its viscosity , water content should be included especially in the frame of changing this parameters dependently form technological temperature used.
Lines: 143-148; 199-200
Thank you for your comment. Suggested information has been added.
10) Subparagraph 2.10- converted or expressed?
Thank you for your comment. The presented method is a calculation. Based on the specifications received from the manufacturers, the nutritional and energy value of the developed high-protein bars was calculated.
11) Line 297. The numbers are not correct, these values are valid for bacteria growth, for yeast and mold average water activity is 0,8 and 0,7 (respectively).
Lines: 324-325
Thank you for your comment. The information on the possibility of the development of microorganisms in a given environment of water activity has been specified.
12) Lines 322-324. Why authors put here sentence about biochemistry of human cells and ATP transformation?
Thank you for your comment. The statement regarding the transformation of ATP was added as a reference to the earlier statements that high-protein bars are a good type of snack, especially for physically active people.
13) How authors refer to the fact that their bars had higher energetic value (370-410 kcal/100g) than most common on the market (250-350 kcal/100g)?
Thank you for your comment. A bar made of WPC + GS proteins was determined as the control sample and the standard most popular on the market and the remaining samples were compared to this variant. The higher energy value compared to market bars may result from the use of a significant amount of fat in the developed recipe. Fat as the nutrient with the highest energy value (9 kcal/g).
14) Line 332-333. this sentence is not necessary, the polyols are not the fiber.
Thank you for your comment. The sentence has been deleted.
15) Line 343. In which way fiber is removing toxins in human body?
Lines: 374-377
Thank you for your comment. Information on the detoxifying effect of fiber has been added.
16) Subparagraphs 3.4 and 3.5. There is lack of explanation what for such measurements were conducted, the aim for it.
Lines: 406-408; 441-444
Thank you for your comment. Suggested information has been added.
17) Line 454. “syrups provide hydrophobic properties similar to those of the standard products.” And what are these standard hydrophobic properties?
Lines: 488-491
Thank you for your comment. The conclusion has been reformulated.
18) There is very few discussion (comparison with other authors) as well as poor explanation of some conducted measurements.
Thank you for your comment. The small number of comparisons to other studies in the case of some analyzes is related to the small amount of literature data available so far taking into account the combinations of proteins and syrups quoted in our study. In addition, studies available in the literature do not usually concern tests performed on industrial infrastructure, using identical types of tests and used ingredients, due to the rather complex composition and high degree of processing.
19) Very often authors are writing “method by [number]”, please correct.
Lines: 281-282; 284; 295; 308; 331-332; 389; 425-426; 433; 436; 451; 457; 470-471
Thank you for your comment. Author records for suggested citations have been changed.

Reviewer 2 Report
The manuscript “Physicochemical, Nutritional, Microstructural, Surface and Sensory Properties of a Model High-Protein Bars Intended for Athletes Depending on the Type of Protein and Syrup Used” is generally very well written and contains data of some relevance for a general readers as well as of high relevance for specialists in the topic. Although the subject of the paper could be of interest for the readers of the journal, the paper needs some corrections:
- Please standardize abbreviations e.g. for protein from soybean the abbreviations “SOY” and “SPI” are used.
- Page 5, lines 223-225 and page 10, lines 354-357: On what basis was there a correlation found? Is it possible to ask for statistical information related to these correlations?
- Page 7, line: 255: In my opinion, if we cite a specific paper by writing "by", then we should provide at least the first author and year, and not only the reference number. Please check the full article.
- Page 8, lines: 301, 302 and page 9, lines 331, 335 and page 12, line 396 and page 13, lines 435, 444: Instead of the word "had" it is better to use the word "was characterized by".
- Page 8, lines: 292, 295: the subscript in aw is missing.
- Page 1, line 25, page 5, line 196, page 8, lines 301, 302: the superscript in cm3 is missing.
- Figure 1. Why was this magnification selected? Did the photos taken at different magnification provide any information?
- Figure 2: In the description of the y axis, the superscript in cm3 is missing.
- Figure 3: In my opinion, separating Figure 3a from 3b with text is not a good solution.
- Were all analyzes of the obtained bars carried out after a 3-week storage period? Unfortunately, this issue is not clear to me. In my opinion, such information should be included in the methodology.
- Marking the best prognostic samples in Figure 2 with a different color is, in my opinion, a very interesting solution.
Author Response
We have corrected our manuscript with regard to Reviewers comments:
Reviewer #1: yellow color
Reviewer #2: green color
Reviewer #3: blue color
If Reviewers had the same comments: pink color
Editorial Office comments: grey color
Reviewer 2 - Review of Manuscript: ijerph-1613239
The manuscript “Physicochemical, Nutritional, Microstructural, Surface and Sensory Properties of a Model High-Protein Bars Intended for Athletes Depending on the Type of Protein and Syrup Used” is generally very well written and contains data of some relevance for a general readers as well as of high relevance for specialists in the topic. Although the subject of the paper could be of interest for the readers of the journal, the paper needs some corrections:
- Please standardize abbreviations e.g. for protein from soybean the abbreviations “SOY” and “SPI” are used.
Pages: 136; 158
Thank you for your comment. The abbreviations has been changed.
- Page 5, lines 223-225 and page 10, lines 354-357: On what basis was there a correlation found? Is it possible to ask for statistical information related to these correlations?
Pages: 246-248; 384-387
Thank you for your comment. The statement was reformulated.
- Page 7, line: 255: In my opinion, if we cite a specific paper by writing "by", then we should provide at least the first author and year, and not only the reference number. Please check the full article.
Pages: 281-282; 284; 295; 308; 331-332; 389; 425-426; 433; 436; 451; 457; 470-471
Thank you for your comment. Citations have been changed.
- Page 8, lines: 301, 302 and page 9, lines 331, 335 and page 12, line 396 and page 13, lines 435, 444: Instead of the word "had" it is better to use the word "was characterized by".
Pages: 327; 329; 361; 421; 428-429; 449; 470
Thank you for your comment. The suggested modifications have been made.
- Page 8, lines: 292, 295: the subscript in awis missing.
Pages: 319
Thank you for your comment. The subscript has been added.
- Page 1, line 25, page 5, line 196, page 8, lines 301, 302: the superscript in cm3is missing.
Pages: 218; 328; 330
Thank you for your comment. The subscript has been added.
- Figure 1. Why was this magnification selected? Did the photos taken at different magnification provide any information?
Thank you for your comment. The reason for choosing the highest possible magnification was due to the most visible differences in the surface structure of the tested types of high-protein bars.
- Figure 2: In the description of the y axis, the superscript in cm3is missing.
Pages: 312
Thank you for your comment. The subscript has been added.
- Figure 3: In my opinion, separating Figure 3a from 3b with text is not a good solution.
Pages: 356-357
Thank you for your comment. The location of the chart has been changed.
- Were all analyzes of the obtained bars carried out after a 3-week storage period? Unfortunately, this issue is not clear to me. In my opinion, such information should be included in the methodology.
Pages: 161-163
Thank you for your comment. Information on the length of the storage period prior to testing has been added.
- Marking the best prognostic samples in Figure 2 with a different color is, in my opinion, a very interesting solution.
Thank you for your comment. Unfortunately, introducing a similar solution to the remaining charts will not be possible due to the type of charts made so far and the large number of data presented on most of them.

Reviewer 3 Report
In this manuscript, MaÅ‚ecki et.al. tested various vegetable protein sources and liquid syrups on the properties of high-protein bars intended for athletes. Various combinations of proteins and syrups were used to determine the textural, physicochemical, nutritional, surface properties, microstructure, sensory parameters, and technological suitability. The authors have written a detailed analysis of each factor and carried out thorough experiments. The authors also showed that how these combinations help reduce the sugar content and increased suitability for storage. The authors investigate use of hydrocolloids like plant protein isolates as alternative to animal proteins and explore vegetable liquid fibers to replace glucose and glucose-fructose syrups. This study provides a plant-based, low sugar, better for storage, fiber-rich high-protein bar model for the athletes and general consumers. Whey protein concentrate (WPC) in combination with glucose syrup (GS) was used as a control and the authors concluded that Soy proteins (SOY) + oligofructose (OF), SOY + Maltitol (ML) and Rice proteins (RPC) + OF may provide the best alternative. It is a well written manuscript and I recommend it for the acceptance after minor changes. In the materials and methods, authors give a reference for the preparation of ‘Preparation of high-protein bars’ in line 138. In addition to the reference, if the authors can add a brief description of the methodology, it would help the general audience. Please specify the number of samples (n=?) for each experiment/figure, preferably in the figure legend. Figure 3 and Figure 6 do not have error bars. Were the experiments performed a single time for these figures? Please comment on the reproducibility of these experiments and if possible add error bars.
Author Response
We have corrected our manuscript with regard to Reviewers comments:
Reviewer #1: yellow color
Reviewer #2: green color
Reviewer #3: blue color
If Reviewers had the same comments: pink color
Editorial Office comments: grey color
Reviewer 3
In this manuscript, MaÅ‚ecki et.al. tested various vegetable protein sources and liquid syrups on the properties of high-protein bars intended for athletes. Various combinations of proteins and syrups were used to determine the textural, physicochemical, nutritional, surface properties, microstructure, sensory parameters, and technological suitability. The authors have written a detailed analysis of each factor and carried out thorough experiments. The authors also showed that how these combinations help reduce the sugar content and increased suitability for storage. The authors investigate use of hydrocolloids like plant protein isolates as alternative to animal proteins and explore vegetable liquid fibers to replace glucose and glucose-fructose syrups. This study provides a plant-based, low sugar, better for storage, fiber-rich high-protein bar model for the athletes and general consumers. Whey protein concentrate (WPC) in combination with glucose syrup (GS) was used as a control and the authors concluded that Soy proteins (SOY) + oligofructose (OF), SOY + Maltitol (ML) and Rice proteins (RPC) + OF may provide the best alternative. It is a well written manuscript and I recommend it for the acceptance after minor changes. In the materials and methods, authors give a reference for the preparation of ‘Preparation of high-protein bars’ in line 138. In addition to the reference, if the authors can add a brief description of the methodology, it would help the general audience. Please specify the number of samples (n=?) for each experiment/figure, preferably in the figure legend. Figure 3 and Figure 6 do not have error bars. Were the experiments performed a single time for these figures? Please comment on the reproducibility of these experiments and if possible add error bars.
Pages: 254; 316-317; 396;
Thank you for your comments. In the case of tests performed in several repetitions, the suggested information (n) has been added in the description of tables and graphs. It is not possible to add error bars for the determinations shown in Figures 3 and 6, because in the case of Figure 3, the test was performed using a computational method, consisting in developing a database of information declared by producers of each ingredient on the quality specification and creating a nutritional value calculation assuming the percentage of individual ingredients based on the developed recipe. The determination of the TSI coefficient presented in Figure 6 was performed in one repetition. In addition, this type of determination consists in a huge number of measurements of the device within a specified time of the analysis and the determination of error bars may affect the illegibility of the graph.

Round 2
Reviewer 1 Report
The manuscript has been improved, authors referred to all raised issues. I recommend this manuscript for publication.
Reviewer 2 Report
Dear Authors,
Thank you for improving the paper according to my suggestions. I have no more comments.
Best regards
Joanna